# Monitoring Patients Reported Outcomes after Valve Replacement Using Wearable Devices: Insights on Feasibility and Capability Study: Feasibility Results

**DOI:** 10.3390/ijerph18137171

**Published:** 2021-07-04

**Authors:** Honoria Ocagli, Giulia Lorenzoni, Corrado Lanera, Alessandro Schiavo, Livio D’Angelo, Alessandro Di Liberti, Laura Besola, Giorgia Cibin, Matteo Martinato, Danila Azzolina, Augusto D’Onofrio, Giuseppe Tarantini, Gino Gerosa, Ester Cabianca, Dario Gregori

**Affiliations:** 1Unit of Biostatistics, Epidemiology and Public Health, Department of Cardiac, Thoracic, Vascular Sciences and Public Health, University of Padova, 35121 Padova, Italy; honoria.ocagli@unipd.it (H.O.); giulia.lorenzoni@unipd.it (G.L.); corrado.lanera@unipd.it (C.L.); matteo.martinato@unipd.it (M.M.); danila.azzolina@uniupo.it (D.A.); 2Department of Cardiac, Thoracic and Vascular Sciences, University of Padua Medical School, 35121 Padua, Italy; alessandro.schiavo.2@gmail.com (A.S.); livio.doc@gmail.com (L.D.); alessandro.diliberti@gmail.com (A.D.L.); giuseppe.tarantini@unipd.it (G.T.); 3Saint Paul’s Hospital, University of British Columbia, Vancouver, BC V6Z 1Y6 VBC, Canada; laura.besola@hotmail.it; 4Cardiac Surgery Unit, Department of Cardiac, Thoracic, Vascular Sciences and Public Health, University of Padova, 35121 Padova, Italy; cibin.gio@gmail.com (G.C.); augusto.donofrio@unipd.it (A.D.); gino.gerosa@unipd.it (G.G.); 5Department of Translational Medicine, University of Piemonte Orientale, 28100 Novara, Italy; 6Cardiology Unit, Dipartimento Strutturale Cardio-vascolare, Azienda ULSS 8 Berica, 36100 Vicenza, Italy; ester.cabianca@aulss8.veneto.it

**Keywords:** surgical aortic valve replacement, transcatheter aortic valve replacement, physical function, wearable devices, feasibility

## Abstract

Wearable devices (WDs) can objectively assess patient-reported outcomes (PROMs) in clinical trials. In this study, the feasibility and acceptability of using commercial WDs in elderly patients undergoing transcatheter aortic valve replacement (TAVR) or surgical aortic valve replacement (SAVR) will be explored. This is a prospective observational study. Participants were trained to use a WD and a smartphone to collect data on their physical activity, rest heart rate and number of hours of sleep. Validated questionnaires were also used to evaluate these outcomes. A technology acceptance questionnaire was used at the end of the follow up. In our participants an overall good compliance in wearing the device (75.1% vs. 79.8%, SAVR vs. TAVR) was assessed. Half of the patients were willing to continue using the device. Perceived ease of use is one of the domains that scored higher in the technology acceptance questionnaire. In this study we observed that the use of a WD is accepted in our frail population for an extended period. Even though commercial WDs are not tailored for clinical research, they can produce useful information on patient behavior, especially when coordinated with intervention tailored to the single patient.

## 1. Introduction

Use of consumer wearable devices (WDs) is increasing both in daily usage and in clinical trials [1]. Fuller et al. [1], in a recent systematic review, showed that various brands of WDs were used in clinical studies, offering accurate measures for steps and heart rate.

The International Data Corporation (IDC) reported a Year-Over-Year Growth of 35.1% in 2020 in the global market of WDs [2]. In clinical setting, the introduction of WDs has brought new challenges to face. At first, the European Medicines Agency did not release any specific guidance addressing the use of WDs. While still recognizing the importance of their use for drugs development, the appropriateness for a specific population, the validity of collected data and the management of large amounts of data collected through the device, issues concerning the choice of the suitable device depending on safety have emerged [3].

A recent US national survey on the use of wearable healthcare devices showed that 82.38% of the people involved in the study are willing to share their health data recorded by their WDs with their healthcare professionals [4]. The use of these technologies tends to decline with advancing age, although the elderly, especially those with a chronic disease, are one of the populations that could benefit most from continuous monitoring in their daily setting. The elderly, however, have poor knowledge on the use of WDs [5]. Various studies have investigated the impact of commercial WDs on the elderly. For example, WDs appear to be accurate in measuring step counts and activity duration in community-dwelling adults [6] and have a positive impact on health as their use had increased physical activity in obese patients [7]. Since WDs are often not tailored for the elderly [8], it is important to evaluate their acceptability in their daily life. A study evaluating the acceptance of WDs among the elderly showed that they seem to accept them and understand the importance of their use in healthcare setting [9].

Patients-reported outcome measures (PROMs) have been suggested in literature to be integrated in clinical trials [10], since clinical outcomes do not measure the patients’ perception of their health status or functional being. The assessment of PROMs (i.e., physical function and quality of life), should be encouraged in elderly patients [11]. Physical function assessment can be useful for evaluating outcomes not directly related to the disease, but still relevant to maintaining personal dependence [12]. WDs can objectively assess physical function in daily clinical practice [13,14].

High levels of physical function are essential for the success of cardiac procedures, so much so that ad hoc cardiac rehabilitation programs have been established to improve patients’ functional recovery through exercise therapy [15]. In the available literature some trials have shown that, for example, after both transcatheter aortic valve replacement (TAVR) and surgical aortic valve replacement (SAVR), physical function is improved. SAVR and TAVR are both highly valuable options for patients with heart failure (HF), a condition that affect mostly elderly patients as showed in a study on time trends in first hospitalization for HF in a community-based population [16]. Patients who had a higher level of physical function before the procedure showed more favorable trajectories [12]. Various studies have compared the two procedures in both high [17,18] and low surgical risk patients [19]. In these studies, New York Heart Association (NYHA) score or, more often self-reporting questionnaire [12,20,21] were used to evaluate physical functions of these patients. However, these instruments have some limitations related to self-reporting, for example in recall and desirability biases.

The present work aims to describe the feasibility of using WDs to monitor PROMs in patients who undergoing TAVR/SAVR enrolled in the run-in phase of the Capability study [22]. In this study, the feasibility and acceptability of using a commercial wearable device in elderly patients undergoing TAVR or SAVR will be explored. 

## 2. Materials and Methods

### 2.1. Study Design and Inclusion Criteria

The study design characteristics and inclusion criteria are described elsewhere [22]. This is a multicenter prospective observational study, that enrolled patients undergoing SAVR and TAVR according to the evaluation of the local heart team since March 2018 at the University hospital of Padova and the hospital of Vicenza.

### 2.2. Data Collection and Procedures

Patients were enrolled at least one week prior to the procedure. Questions related to socio-demographic characteristics, risk factors, physical activity and clinical characteristics were the information collected at baseline assessment. WDs along with the smartphone were delivered at baseline assessment. Participants completed the same assessments, except for demographic characteristics, at one month, three months, six months and twelve months after the procedure. From March 2020 to the end of the follow up period, assessments were carried out only by telephone given the restrictions related to the COVID-19 pandemic.

### 2.3. Ethical Considerations

The study is registered in Clinicaltrials.gov (NCT03843320) and approved by the hospital ethics committee with the protocol No 943 (4 January 2019). Written informed consent was obtained from all patients for study participation and for data collection through their Garmin© device account after proper explanation of the study outcomes by a physician.

### 2.4. Device

A Garmin^©^ Vívoactive^®^ 3 smartwatch activity tracker device and a smartphone with the Garmin Connect^©^ application (GARMIN, Olathe, KS, U.S.) installed in the smartphone for data transfer were provided to each patient at baseline assessment. The device used in the study was commercially available and was chosen for its ability to estimate steps count and sleep duration in free living environments both in the general and elderly population [23]. Both the patient and his/her caregiver were provided with information and trained on the use of the WD and the smartphone. 

### 2.5. Device Setup and Usage

A personal account of the Garmin Connect^©^ application has been created for each patient. The app contains only patient demographics (i.e., gender, age, weight, height, wrist of usage of the WD). Both the participants and their caregivers were not granted access to the account. All notifications were disabled in the device and the smartphone was cleaned by all the unrelated applications, to avoid affecting behaviors of the participants. Participants were asked to wear the device on their wrist 24 h a day, including while showering and sleeping, except while charging. They were asked to charge the device daily and sync data weekly. A member of the study assisted the patients at each of follow up for connectivity issues and for collecting data, while also remaining available by phone and possibly in person to troubleshoot the device at any time. 

Patients were also informed that they would be asked to return the device at the end of the study. However, at the last follow up, participants were asked to choose whether to continue using the device with the smartphone for private use. After the end of the study, we collected no further data from the device. 

### 2.6. Measurements

Physical function was assessed through a series of standard and validated tests: Duke Activity Status Index (DASI) [24], Activity of Daily Living (Barthel Index) and Instrumental Activity of Daily Living (IADL)). The DASI is a measure of functional capacity that can be obtained by self-administered questionnaire and already used in patients that underwent TAVR [25]. The Barthel Index (BI) evaluates activity of daily living [26] and has been used in aortic valve replacement [27]. The DASI score ranges from 0 to 58.2, Barthel Index ranges from 0 to 100, for both the instrument, the higher the score, the higher the functional status. IADLs were evaluated with the scale of Lawton and Brody [28]. The higher the score, the greater the person’s abilities. The Epworth sleepiness scale (ESS) was used to evaluate the “subject’s general level of daytime sleepiness” [29], a higher score means higher sleepiness and could be interpreted as follows: 0–5 lower normal daytime sleepiness, 6–10 higher normal daytime sleepiness, 11–12 mild excessive daytime sleepiness, 13–15 moderate excessive daytime sleepiness, 16–24 severe excessive daytime sleepiness [30].

### 2.7. Acceptance of the Technology

Compliance with the use of the WD (i.e., the time the device was worn) was determined by calculating the proportion of days the device was active and the total number of days before and after the procedure. At the end of the follow up, patients were given a questionnaire on technology acceptance, based on the work of Puri et al. [9]. The instrument investigated six key dimensions for WD acceptance: perceived usefulness, perceived ease of use, privacy concerns, perceived risks, facilitating conditions and equipment characteristics. Device acceptance was assessed with the question L33 as suggested in the validation study [9], “Would you use the device you used during the last year to continue to monitor or track your physical activity or health?”

### 2.8. Statistical Analysis

Continuous variables were reported as I, II (median) and III quartiles, categorical variables were presented as absolute numbers and percentages. Wilcoxon’s test and Chi-squared test were used to evaluate differences between TAVR and SAVR, respectively, for continuous and categorical variables. The compliance on the use of the WDs was summarized computing the number of days before the procedure divided by the number of days that data was synchronized in that period, the same was carried out for the follow up period. Significance was evaluated for *p*-value lower than 0.05. Data analysis was performed with R software (version 4.0.3) [31]. 

## 3. Results

### 3.1. Baseline Characteristics

The patients considered eligible for the study were 17, 12 TAVR and 5 SAVR, 4 from the center of Vicenza and the remaining from the University hospital of Padova. Eight patients completed the entire follow up period (Flowchart Figure 1) with an enrolment rate of 47%. Four patients were found to be ineligible after completion of the baseline assessment. 

Table 1 reports patients baseline demographic characteristics by type of procedure, TAVR or SAVR. The overall sample had a median age of 79 years, 78 in the SAVR group and 82 in the TAVR group (*p*-value = 0.046). 

Except for age, there were no statistically differences between the SAVR and the TAVR groups. Patients were mainly female (10, 59%), married or cohabiting (12, 71%). Hypertension was the main risk factors in both groups, (14, 82%). Only two patients underwent TAVR procedure in urgent status. Patients, according to clinical frailty scale were more than vulnerable in the TAVR group (4, 33%; vulnerable, 3, 25% mildly frail and, 2, 17% moderate frail). Ejection fraction was similar in the two groups, median of 60 and 58, respectively, in SAVR and TAVR (*p*-value 0.49). 

### 3.2. Score Trend

Table 2 reports the trend of the Barthel Index, DASI score, IADL index and ESS score at each follow up according to the type of procedure. SAVR patients reported higher physical function levels than TAVR especially at 12-month follow up (BI 100 vs. 85, DASI 19.9 vs. 12.8, respectively, for SAVR and TAVR). TAVR recorded lower levels of physical function at baseline according to the DASI score (30.4 vs. 14.4). Physical function level was similar at 12 months of follow up compared to the baseline assessment in both groups according to BI. The DASI score instead showed a decrease for both groups: 30.4 vs. 19.9 and 14.4 vs. 12.8, respectively, for SAVR and TAVR. As for the IADL, there was a decrease at 12 months from baseline both for TAVR and SAVR (6 vs. 4 for SAVR, 6 vs. 5 for TAVR). The ESS score was highest at 12 months follow up for both SAVR and TAVR, both groups had lower normal daytime sleepiness at each follow up.

### 3.3. Device’s Data

Figure 2, Figure 3 and Figure 4 show the daily trend for each patient of rest heart rate, the number of steps and the number of hours of sleep, respectively, recorded by the device divided by baseline and follow up period and according to the procedure. Patients that underwent TAVR had a median rest heart rate higher than SAVR in the follow up period (median 64 vs. 57) (Figure 2). 

As for the number of steps, the SAVR group seems more active, median of 2701 steps/day vs. 1735 of TAVR (Figure 3).

Regarding the number of hours of sleep, both groups after the procedure had a similar median number of hours of sleep (7.3 SAVR vs. 7.0 TAVR) (Figure 4). 

### 3.4. Compliance and Acceptance of the Technology

Data were mainly synchronized by the caregivers in both groups, only one TAVR patient synchronized by himself the device. The WDs and the smartphones were charged mainly by the caregivers, two TAVR patients charged their devices themselves. None of the patients or their caregivers had used a wearable device prior to the study. The compliance, evaluated as the percentage of days device use divided by the total number of days, was similar in both groups and reached a median of 75.1% in SAVR and 79.8% in TAVR (Table 3). 

Figure 5 shows an overview of WDs wear time for the entire study period. Valid and missing data days are showed for each participant. The percentages of missing data vary from 0, no data, up to 70.15%. Noticeably, the patients with only 0.55% and 0% of missing data were those who could rely on the daily assistance of the caregiver. Only one patient collected less than 30% of data: in this case the caregiver was not often in contact with the patient and the patient was not able to use the smartphone autonomously. 

One patient, not reported in the analysis, stopped using the activity tracker before the surgery due to the tightness of the strap and the caregiver unwillingness to synchronize data. Missing data varied between 11.5% and 32.88%: in these cases, caregivers were available on a weekly basis. The patient who collected data by himself eventually reduced his compliance due to the long waiting-time for the surgery. In the preoperative period, SAVR patients synchronized data more often than TAVR ones, however this result could be affected by the difference in the length of the pre-operative period, which was longer for SAVR patients. In the follow up period, the compliance was similar on both groups, 75.1% vs. 75.9% for SAVR and TAVR, respectively.

### 3.5. Technology Acceptance Questionnaire

Table 4 reports the technology acceptance questionnaire scores for each of the seven dimensions based on the type of procedure. Except for perceived risk dimension (median 10 vs. 5.5 SAVR vs. TAVR, *p*-value = 0.016) there were no differences in all the dimensions among the two groups. The perceived risk dimensions score was higher in patients that underwent SAVR. Four patients reported that they would wish to continue to use the device (3 TAVR, 1 SAVR). The device characteristics satisfied the patients, the equipment characteristics reached for both group the maximum score, the same for perceived ease of use, also perceived usefulness showed good levels. Conversely, privacy concern (median 8 vs. 9, SAVR vs. TAVR), perceived risk (10 vs. 5 SAVR vs. TAVR) and subjective norm (10 vs. 9, SAVR vs. TAVR) showed lower level of satisfaction.

## 4. Discussion

This study explores the feasibility and acceptability of using a commercial wearable device in elderly patients undergoing TAVR or SAVR. 

The main findings of our study showed that there was an overall good compliance in wearing the device (75.1% vs. 79.8%, SAVR vs. TAVR) with half of the patients willing to continue using the device. Physical function decreased after both SAVR and TAVR, more in TAVR patients. The ESS score increased after the procedure but remained as “normal daytime sleepiness” according to the score, the WDs showed an increase in the number of hours slept up to 7 h per night for both groups.

Our results showed a good overall acceptance of wearing the device during the follow up period. These results are in line with what was reported in a recent literature review which showed a high-level adherence in long term daily use [32]. The acceptance rate in our study was high compared to other studies using the same device [33]. Our results considered a longer follow up period. Studies in the literature, in order to evaluate the compliance and acceptance of a device, considered shorter periods ranging from a few days [34] to weeks [5]. The number of valid days in the follow up period was lower than in the study of Henriksen et al. [35] (265 vs. 292). This result is promising in our population as data synchronization was usually performed by a caregiver. However, this high compliance can be explained by the desire of the patients to use of the WD to contribute to the study. 

The main discomfort reported by our patients were related to the need to use another smartphone to synchronize data with the website and connectivity issues, as also reported in a review of activity trackers for senior citizens [36]. This could easily be avoided by allowing the patients or the caregiver to download the app directly to their own smartphone. Other technical problems were related to changing accidentally the setup of the WD or of the smartphone or forgetting to charge them. In one case, the main problem was related to the unavailability of a WI-FI connection at home. To solve these problems the presence of a technically skilled staff member available on request has proven to be a key point. Other studies showed that technical problems could reduce compliance in the use of WDs [37]. 

There is a growing interest in evaluating the acceptability of wearable devices both in the general and elderly population. A greater acceptance of these devices can improve the quality of real time data collection [38]. Various studies have shown that older adults accept the use of wearable devices, especially after facing acceptance barriers [9,35,39]. Perceived ease of use is one of the domains that has higher level in our results. This is in line with the findings of a recent study [40] and related to the fact that, while technology use has been increasing also in elderly people, they still need additional information and support to adopt it [41]. A recent study showed that commercial wearable devices are reliable for measuring physical activity level in elderly patients in real-life setting [42]. Despite this increase and the fact that elderly population is the one that can benefit the most from the use of these devices, especially when chronic illnesses are present [34], very few elderlies currently use daily WDs [5] or consider them in health monitoring [43]. However, the perceived ease of use recorded with the questionnaire was high for both groups, respectively, median of 30 for SAVR and 32 points for TAVR (maximum 35). 

Even though the compliance was high, the perceived usefulness was not as high in both groups, median of 16 and 17, respectively, for SAVR and TAVR (64% and 68% of the overall score). This is likely sue to the fact that most of the device-related procedures were performed by caregivers and not by patients. During the follow up encounters, the researcher reinforced the importance of collecting data from the activity tracker. Caregivers, on the other hand, reported that having the possibility to see heart rate, number of steps and number of hours of sleep was useful for obtaining information on the health of the patient. 

Participants when asked How much would you be willing to pay for the device you wore during the last year? (question L35) answered mainly 0 euro (4, 50%), 2 from 1 to 50 euros and 1 from 51 to 100 euros. This contrasts with what Kekade [5] reported. This may be related to the fact that in our study the population was extremely elderly (median 78 and 82, respectively, for SAVR and TAVR), while the Kekade study considered elderly over 65 years of age. 

Cardiac rehabilitation after both TAVR and SAVR have shown improved functional capacity in a recent review [27]. A reduction in physical function level both according to BI and DASI score from baseline to one year follow up was showed in our sample. This result is in contradiction of what reported in other metanalysis for TAVR [21] and both TAVR and SAVR [12]. Cardiac rehabilitation was required only for two participants, both underwent SAVR. So, the sole adoption of the wearable devices is not enough to improve physical activity in these patients. In the future it would be useful to help patients in recognizing the long-term benefits of the device, along with social support as suggested by Kononova [44] in his study on tracker perceptions among older adults. Moreover, it would be helpful to use the commercial wearable activity tracker in a broader physical activity intervention as shown by the review of Brickwood [45].

Our data showed that patients that underwent SAVR were more active than TAVR patients. A functional decline or lack of improvement after the procedure was found, as already found by Kim in his study evaluating changes in functional status in the year after aortic valve replacement [12]. These differences between SAVR and TAVR and no change in functional status for TAVR may be related to the fact that TAVR group had a higher median age and none of them had rehabilitation after the procedure. 

### Limitations

The main limitations of this study derived from the difficulties to recruit patients with our strictly inclusion criteria. Moreover, the study design was adapted to the limitations derived from the spread of the pandemic of COVID-19. 

## 5. Conclusions

Given the importance of developing a proper observational study to evaluate the use of wearable devices in elderly patients that underwent TAVR or SAVR, our feasibility study provided useful insights on how to implement further our project. In this study we have observed that the use of a WD is accepted in our frail population for an extended period. The use of a WD for collecting data allows the collection of data on daily basis, directly at home, with improved quality since data does not have to be manually entered and checked. However, the use of WDs in clinical trials requires an additional effort on behalf of the research team. A researcher must be available to set up the device at the beginning of the study and to solve problems related to the device. The collection of data through a WD requires always to have an application for collecting data: this could cause problems related to connectivity and device communication. Moreover, the transmission of data requires a minimum ability in the use of technologies in participants. The help of a caregiver is required especially when participants are elderly and sometimes this is not possible. 

However, the presence of the device alone is not enough to encourage healthy behaviors. Therefore, it would be useful to create a coordinated intervention with a physiatrist to implement a physical activity program tailored to the single patient. As also reported in the literature, elderly people need more training session than young people, this is particularly true in population such as ours, who are not used to any type of electronic device. Considering that older people, especially frail ones, often rely on the help of a caregiver, they also need to be trained in the potential of these tools. Another suggestion could be the creation of custom reports to allow patients to view their progress in terms of physical activity. In our experience it would be faster to download the app directly on the smartphone of patients or assistants to avoid connectivity problems.

Even though commercial wearable devices were not tailored for clinical research, they could produce useful information on the behavior of the patient. Furthermore, the implementation of the use of these devices, especially in elderly, will minimize the need to attend the clinical study center, with the potential of reducing the time and costs related to person-by-person visits. 

## Figures and Tables

**Figure 1 ijerph-18-07171-f001:**
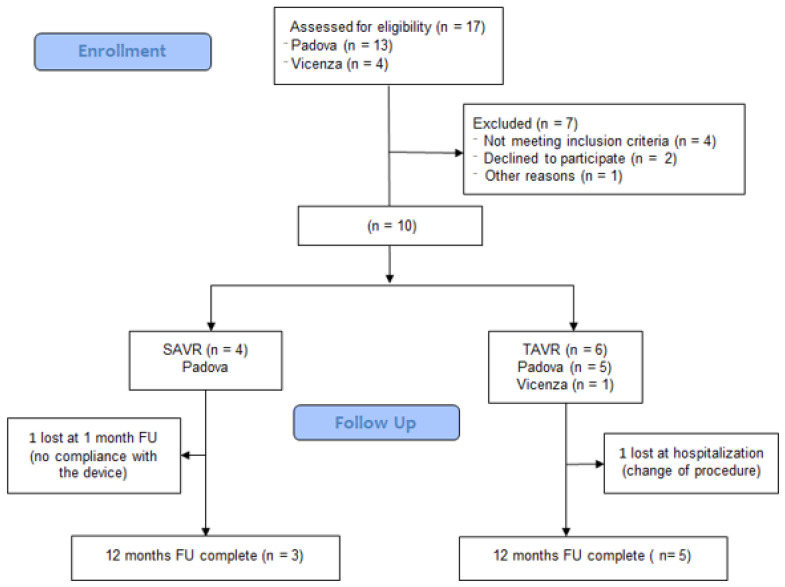
Flowchart of eligible patients in the two centers, Padova and Vicenza.

**Figure 2 ijerph-18-07171-f002:**
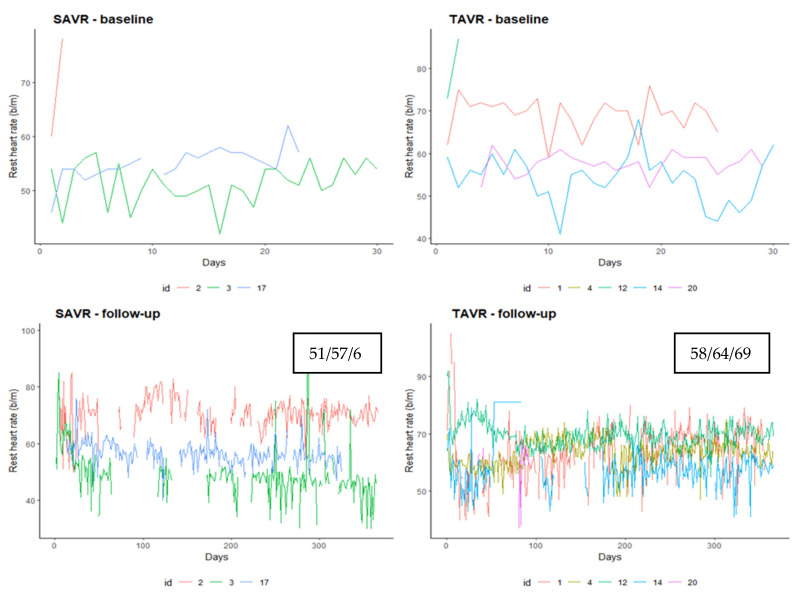
Rest heart rate trend recorded by the device according to procedure and baseline vs. follow-up period. Data in the box are the I, median and III quartile of rest heart rate.

**Figure 3 ijerph-18-07171-f003:**
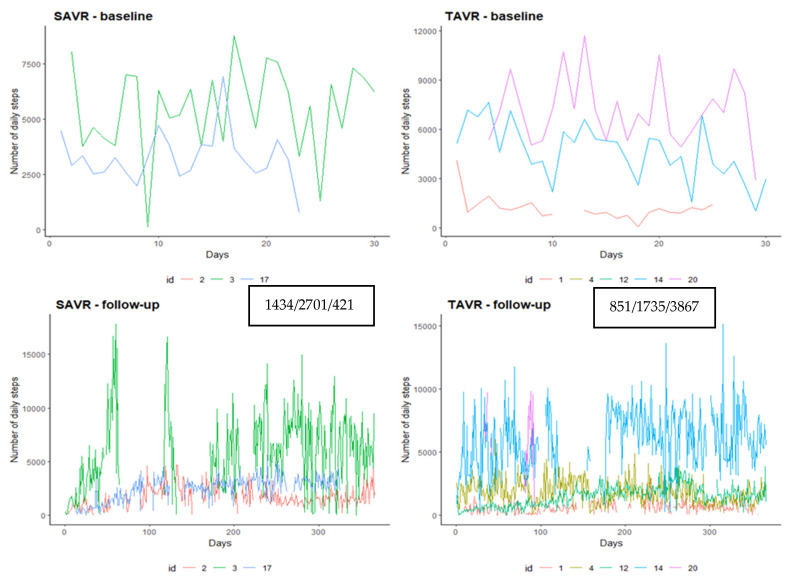
Number of steps trend recorded by the device according to procedure and baseline vs. follow-up period. Data in the box are the I, median and III quartile of number of steps.

**Figure 4 ijerph-18-07171-f004:**
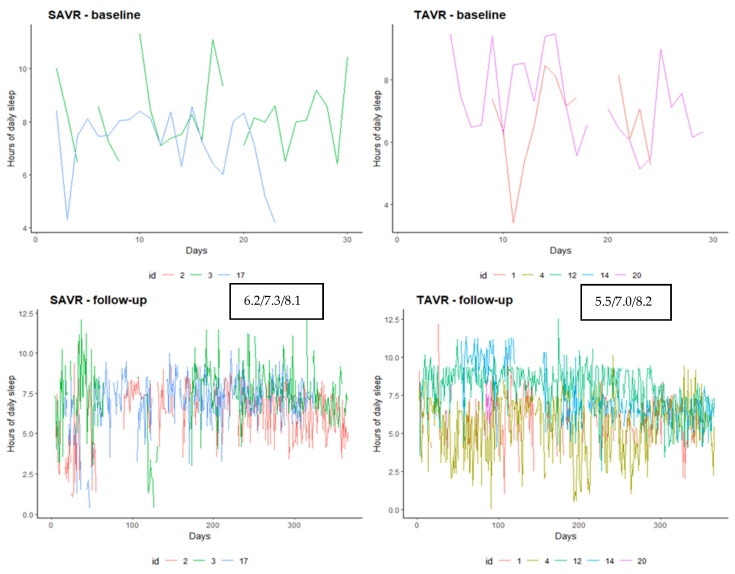
Number of hours slept trend recorded by the device according to procedure and baseline vs. follow-up period. Data in the box are the I, median and III quartile of hours of sleep.

**Figure 5 ijerph-18-07171-f005:**
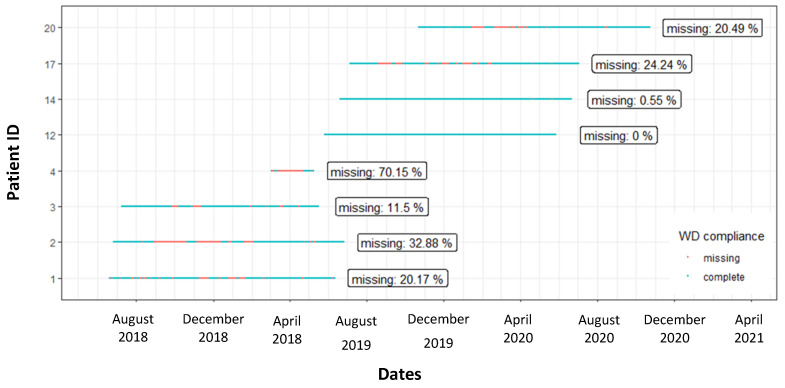
Wearable device wear time for the entire follow-up period for each participant. A line that represents the period of enrolment is reported for each participant, in blue are reported the days without missing data, in red the days with missing data.

**Table 1 ijerph-18-07171-t001:** Baseline characteristics of the sample according to the type of intervention, surgical aortic valve replacement (SAVR) or transcatheter aortic valve replacement (TAVR).

		N	SAVR (N = 5)	TAVR (N = 12)	Overall (N = 17)	*p* Value
Center	Padova	17	100% (5)	67% (8)	76% (13)	0.14
	Vicenza		0% (0)	33% (4)	24% (4)	
Drop out	Yes	17	40% (2)	50% (6)	47% (8)	0.71
Gender	Female	17	40% (2)	67% (8)	59% (10)	0.31
	Male		60% (3)	33% (4)	41% (7)	
Age		17	76/78/79	79/82/85	78/79/83	0.046
Marital status	Married cohabitant	17	80% (4)	67% (8)	71% (12)	0.58
	Widowed unmarried		20% (1)	33% (4)	29% (5)	
Educational level	Primary	16	40% (2)	64% (7)	56% (9)	0.38
	Secondary		60% (3)	36% (4)	44% (7)	
Risk factors	Diabetes	17	0% (0)	8% (1)	6% (1)	0.47
	Hypertension		100% (5)	75% (9)	82% (14)	
	Smoker		0% (0)	17% (2)	12% (2)	
Status	Elective	16	100% (5)	82% (9)	88% (14)	0.31
	Urgent		0% (0)	18% (2)	12% (2)	
Clinical frailty scale	Well	17	40% (2)	8% (1)	18% (3)	0.25
	Managing well		40% (2)	17% (2)	24% (4)	
	Vulnerable		0% (0)	33% (4)	24% (4)	
	Mildly Frail		20% (1)	25% (3)	24% (4)	
	Moderate Frail		0% (0)	17% (2)	12% (2)	
NYHA class	1	16	20% (1)	9% (1)	12% (2)	0.82
	2		60% (3)	64% (7)	62% (10)	
	3		20% (1)	27% (3)	25% (4)	
COPD		17	0% (0)	8% (1)	6% (1)	0.78
Ejection fraction		15	57/60/62	50/58/61	54/58/62	0.49

Abbreviations: NYHA = New York heart association; COPD = Chronic obstructive pulmonary disease.

**Table 2 ijerph-18-07171-t002:** Baseline characteristics according to the type of intervention. Data are reported for that underwent SAVR (3 patients) and TAVR (5 patients).

		Baseline	1 Month	3 Months	6 Months	12 Months
Barthel Index	SAVR	88/95/98	92/95/98	95/100/100	72/95/98	80/100/100
	TAVR	90/90/100	70/70/85	80/85/90	90/95/95	85/85/100
DASI score	SAVR	21.6/30.4/31.9	8.2/16.4/20.8	14.5/16.2/19.8	12.6/23.4/25.2	12.2/19.9/22.6
	TAVR	10.7/14.4/20.4	7.2/7.2/14.4	12.7/16.4/32.5	10.7/15.4/26.9	7.2/12.8/24.4
IADL score	SAVR	4.5/6.0/6.5	2.5/5.0/6.0	2.5/3.0/4.0	1.5/3.0/3.5	3.5/5.0/5.5
	TAVR	5/6/6	2/4/4	2/4/6	2/3/6	3/5/8
ESS score	SAVR	1.0/2.0/6.0	2.5/4.0/5.0	2.5/3.0/4.5	3.0/4.0/6.0	3.0/3.0/5.5
	TAVR	3/3/3	3/3/4	3/5/5	4/4/5	5/5/5

Abbreviations: SAVR = Surgical aortic valve replacement; TAVR = transcatheter aortic valve replacement.

**Table 3 ijerph-18-07171-t003:** Compliance on the use of the device according to the procedure, SAVR vs. TAVR. Data are reported as I quartile, median and III quartile, median and standard deviations.

Period.	SAVR (N = 3)	TAVR (N = 5)	Combined (N = 8)	*p* Value
Overall	76/82/96 84+/13	39/83/100 71+/34	73/82/100 76+/28	0.92
Pre	89/100/100 93+/12	25/86/100 67+/39	65/93/100 77+/33	0.47
Post	71.1/75.1/80.1 75.8+/0.091	79.5/79.8/99.5 75.9+/32.4	73.1/79.6/88.8 75.9+/25.0	0.5

Abbreviations: SAVR = Surgical aortic valve replacement; TAVR = transcatheter aortic valve replacement.

**Table 4 ijerph-18-07171-t004:** Differences in the technology acceptance questionnaire scores per dimension according to the procedure for the overall sample. For each dimension is reported the maximum score. Data are reported as I, II and III quartiles. In parenthesis is reported the highest score for each dimension.

	N	SAVR (N = 3)	TAVR (N = 5)	Combined (N = 8)	*p* Value
Perceived usefulness (30)	7	16/16/16	17/17/17	17/17/17	0.16
Perceived ease of use (35)	7	29/30/31	32/32/32	30/32/32	0.57
Equipment characteristics (10)	7	30/31/32	30/31/32	30/31/32	0.73
Privacy concern (15)	7	7.0/8.0/9.0	9.0/9.0/9.0	8.0/9.0/9.5	0.72
Perceived risk (15)	7	10/10/10	5/5/5	5/5/8	0.009
Facilitating conditions (10)	7	5.8/6.5/7.2	7.0/7.0/7.0	6.5/7.0/7.5	0.86
Subjective norm (15)	7	9.5/10.0/10.5	8.0/9.0/10.0	8.5/9.0/10.5	0.48

Abbreviations: SAVR = Surgical aortic valve replacement; TAVR = transcatheter aortic valve replacement.

## Data Availability

The data presented in this study are available on request from the corresponding author. The data are not publicly available due to privacy restrictions.

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
