# Peer review of "Monitoring Patients Reported Outcomes after Valve Replacement Using Wearable Devices: Insights on Feasibility and Capability Study: Feasibility Results"

_ijerph, 2021, doi:10.3390/ijerph18137171_

Round 1

Reviewer 1 Report

The manuscript describes the feasibility and acceptability of using commercial WDs in elderly patients undergoing trans catheter aortic valve replacement (TAVR) or surgical aortic valve replacement (SAVR) will be explored. This research includes data collection, validated questionnaires and statistical analysis, etc. Experimental results show that the use of a WD is accepted in our frail population for an extended period. However the following issues should be addressed before it could be published.

Detailed Comments:

  1. There are more spaces at the beginning of line 45.
  2. Title 2.3 and 2.5-2.8 have format error and should not be bold.
  3. Title 6.1 on line 336 should not appear here.
  4. Space is required at the beginning of line 265.
  5. The reference on line 277 is incorrectly marked.
  6. Please explain the meaning of the number in the text box in Figure 2, 3, 4.

Author Response

We thank the reviewer for the careful consideration and overall positive judgement given to our work.

Detailed Comments:

There are more spaces at the beginning of line 45.

Modified.

Title 2.3 and 2.5-2.8 have format error and should not be bold.

Modified.

Title 6.1 on line 336 should not appear here.

Modified
Space is required at the beginning of line 265.

Modified
The reference on line 277 is incorrectly marked.

Modified.
Please explain the meaning of the number in the text box in Figure 2, 3, 4.

Thanks for the suggestion. Modified.

Reviewer 2 Report

This is a well-articulated research article. The authors presented a feasibility study on the use of a wearable device to evaluate the long-term outcomes of surgical interventions such as TAVR and SAVR. I have no major comments.

Minor comments:

Line 28: Half of the patients were willing to continue using the device.  

Author Response

Reviewer 2:

This is a well-articulated research article. The authors presented a feasibility study on the use of a wearable device to evaluate the long-term outcomes of surgical interventions such as TAVR and SAVR. I have no major comments.

We thank the reviewer for the careful consideration and overall positive judgement given to our work.

Minor comments:

Line 28: Half of the patients were willing to continue using the device.  

Thanks for the suggestion, modified.